# A Biopsy-Controlled Prospective Study of Contrast-Enhancing Diffuse Glioma Infiltration Based on FET-PET and FLAIR

**DOI:** 10.3390/cancers16071265

**Published:** 2024-03-24

**Authors:** Maciej Harat, Izabela Miechowicz, Józefina Rakowska, Izabela Zarębska, Bogdan Małkowski

**Affiliations:** 1Department of Neurooncology and Radiosurgery, Franciszek Lukaszczyk Oncology Center, 85-796 Bydgoszcz, Poland; 2Department of Clinical Medicine, Faculty of Medicine, University of Science and Technology, 85-796 Bydgoszcz, Poland; 3Department of Computer Science and Statistics, Poznan University of Medical Sciences, 61-701 Poznań, Poland; iza@ump.edu.pl; 4Department of Neurosurgery, 10th Military Research Hospital, 85-681 Bydgoszcz, Poland; jozefina.rakowska@gmail.com; 5Department of Radiotherapy, Franciszek Lukaszczyk Oncology Center, 85-796 Bydgoszcz, Poland; zarebskai@co.bydgoszcz.pl; 6Department of Nuclear Medicine, Franciszek Lukaszczyk Oncology Center, 85-796 Bydgoszcz, Poland; 7Department of Diagnostic Imaging, Ludwik Rydygier Collegium Medicum, Nicolaus Copernicus University, 85-067 Bydgoszcz, Poland

**Keywords:** glioma, glioblastoma, high-grade glioma, biopsy, FET-PET, MRI

## Abstract

**Simple Summary:**

Currently, contrast-enhancing gliomas FLAIRectomies or supramarginal gliresections are gaining momentum. This paper presents a semiquantitative analysis of FET uptake in biopsy targets inside and outside glioma masses based on PET/MR images. An exact threshold to differentiate glioma and astrogliosis within FLAIR using dual-timepoint PET acquisition and referring to various anatomical structures is a major strength of this study. Doing so paves the way for an optimized PET protocol to enable precise and reproducible FET-PET quantification for glioma mapping in clinical trials and practice.

**Abstract:**

Accurately defining glioma infiltration is crucial for optimizing radiotherapy and surgery, but glioma infiltration is heterogeneous and MRI imperfectly defines the tumor extent. Currently, it is impossible to determine the tumor infiltration gradient within a FLAIR signal. O-(2-[18F]fluoroethyl)-L-tyrosine (FET)-PET often reveals high-grade glioma infiltration beyond contrast-enhancing areas on MRI. Here, we studied FET uptake dynamics in tumor and normal brain structures by dual-timepoint (10 min and 40–60 min post-injection) acquisition to optimize analysis protocols for defining glioma infiltration. Over 300 serial stereotactic biopsies from 23 patients (mean age 47, 12 female/11 male) of diffuse contrast-enhancing gliomas were taken from areas inside and outside contrast enhancement or outside the FET hotspot but inside FLAIR. The final diagnosis was G4 in 11, grade 3 in 10, and grade 2 in 2 patients. The target-to-background (TBRs) ratios and standardized uptake values (SUVs) were calculated in areas used for biopsy planning and in background structures. The optimal method and threshold values were determined to find a preferred strategy for defining glioma infiltration. Standard thresholding (1.6× uptake in the contralateral brain) in standard acquisition PET images differentiated a tumor of any grade from astrogliosis, although the uptake in astrogliosis and grade 2 glioma was similar. Analyzing an optimal strategy for infiltration volume definition astrogliosis could be accurately differentiated from tumor samples using a choroid plexus as a background. Early acquisition improved the AUC in many cases, especially within FLAIR, from 56% to 90% sensitivity and 41% to 61% specificity (standard TBR 1.6 vs. early TBR plexus). The current FET-PET evaluation protocols for contrast-enhancing gliomas are limited, especially at the tumor border where grade 2 tumor and astrogliosis have similar uptake, but using choroid plexus uptake in early acquisitions as a background, we can precisely define a tumor within FLAIR that was outside of the scope of current FET-PET protocols.

## 1. Introduction

Contrast-enhanced magnetic resonance imaging (MRI) is the diagnostic tool of choice for gliomas [1]. Diffuse gliomas are characterized by their extensive infiltration of normal brain tissue, but this infiltration is often difficult to see in standard MRI sequences [2,3]. Failing to accurately delineate glioma infiltration could have a negative impact on treatment outcomes, and indeed most treatment failures are local, suggesting incomplete local control [2,3,4]. 

T1-weighted MRI with gadolinium contrast (T1-GAD) is usually used to define gross tumor volume [5]. While T1-GAD sequences are helpful for indicating areas with a compromised blood–brain barrier due to tumor infiltration, infiltrating tumor cells are still often found several centimeters away from the contrast-enhancing mass [6]. Other MRI modalities such as fluid-attenuated inversion recovery (FLAIR), MR spectroscopy (MRS), or apparent diffusion coefficient (ADC) maps may identify infiltrative glioblastoma, but their routine application has been limited due to poor specificity [7], resolution [8], or poor correlation with the highest grade parts [9]. Despite recognition of the limitations of MRI, MRI-guided supratotal resections (i.e., resections beyond any visible MRI abnormalities) are currently not recommended due to the poor supportive evidence base [10]. 

O-(2-[^18^F]fluoroethyl)-L-tyrosine positron emission tomography (^18^F-FET-PET) is a valuable tool for imaging glioma, defining biopsy sites, monitoring treatment responses, and differentiating treatment-related changes from glioma progression [11]. FET-PET is also a promising modality for estimating tumor margins because tracer uptake is independent of blood–brain barrier disruption and allows for the identification of glioma tissue not associated with contrast enhancement on MRI [11,12]. There is evidence that the complete resection of areas with increased amino acid PET tracer uptake prolongs the survival of high-grade glioma (HGG) patients [13,14]. However, work is still required to optimize FET-PET-based tumor volumes to ensure that tumor is distinguished from astrogliosis.

To successfully implement FET-PET for glioma mapping into clinical trials and practice, a standard, optimized PET protocol is urgently needed to enable precise and reproducible FET-PET quantification. Therefore, the primary aim of this study was to differentiate tumor tissue from non-tumoral based on PET/MR images. The secondary aim was to optimize acquisition and analysis protocols for defining glioma infiltration outside the FET hotspot and inside FLAIR. To achieve these aims, we used not only dual-timepoint acquisitions but also tested an uptake in a variety of control tissues to obtain optimized thresholds.

We hypothesized that an optimal methodology and threshold values could be determined to accurately distinguish glioma infiltration from astrogliosis and normal brain structures.

## 2. Methods

### 2.1. Ethical Approval and Consent

The university ethics committee approved the study protocol [KB 647/2015, 20.10.2015]. All subjects gave written informed consent for study participation.

### 2.2. Patients and Inclusion and Exclusion Criteria

This was a prospective study of 23 patients (12 women and 11 men; mean age 47.7 years, range 18–86; mean KPS 90) attending the Department of Neurosurgery of the 10th Military Clinical Hospital, Bydgoszcz, Poland. The details of this cohort can be found in our previous publication [15]. The patients were diagnosed with CNS gliomas based on MRI and clinical assessment. All the patients qualified for stereotactic biopsy. The reasons for CNS diagnostics in these patients included headaches, epileptic seizures, cerebellar syndrome, speech disorders, and limb paresis. In a few patients, the tumor was detected incidentally during diagnostics performed for head injury. 

The inclusion criteria were age > 18 years, suspicion of diffuse adult-type glioma on MRI, and good performance status allowing for surgery. The exclusion criteria were age < 18 years, poor quality of life (KPS < 60), pregnancy, coagulation disorders, no increased FET uptake, and no contrast enhancement present in T1-weighted MRI sequences. 

All the patients underwent serial biopsy from the tumor and peritumoral areas based on hybrid FET-PET/MRI. In total, 308 samples were taken from 23 patients, but samples containing a blood clot or that were uninterpretable were excluded (*n* = 24). An experienced neuropathologist evaluated 284 biopsy samples. A representative trajectory targeting different SUVs at the tumor border in early and late PET and MRI sequences are shown in Figure 1A.

### 2.3. FET-PET Acquisition and Evaluation

Amino acid FET was produced and administered as per international standards [16]. All PET examinations were performed using a hybrid PET/MRI scanner (Siemens Biograph mMR 3T, Siemens AG, Muenchen, Germany) with dual-timepoint acquisition: an early acquisition 5–15 min after radionuclide injection (a.r.i.) and a standard acquisition 40–60 min a.r.i. According to brain tumor imaging guidelines for labeled amino acid analogues [17], patients fasted for at least 4 h before PET measurements. The PET/MR sequences were as in our previous publication [15]. 

All PET images with MRI were co-registered using a commercial fusion algorithm (Brainlab iPlan Image Fusion, Brainlab AG, Munich, Germany) for brain tumor biopsy planning verified by two experienced neurosurgeons. The procedure is described in detail in [15]. Serial biopsies were taken from the FLAIR margins to the central part of the lesion. The FLAIR margins were defined visually by an experienced neurosurgeon as areas without contrast enhancement and hyperintense in FLAIR. The SUVs in other trajectories and targets were defined as follows: Trajectory 1: T1-GAD^+^/PET^+^ (T1-GAD); Trajectory 2: T1-GAD^−^/PET^+^ (PET); Trajectory 3: T1-GAD^+^/PET^−^ (PET^−^); and Trajectory 4: FLAIR/PET^−^ (FLAIR). Trajectory 3 (PET^−^) represented areas of low-to-moderate uptake outside a hotspot relative to uptake in the contralateral brain.

All the SUVs and target-to-background (TBR) values were defined for early and standard (late) acquisitions. The TBR was expressed as the ratio of target uptake to uptake in the reference contralateral brain or other structures, as defined below. The mean and maximum TBRs (TBR_mean_ and TBR_max_, respectively) were calculated by dividing the point TBR within the target to the mean or maximum TBR in the reference area. An SUV was defined in each target based on the point values taken from voxels at the site of biopsy, as in iPlan Stereotaxy (BrainLAB, AG, Munich, Germany). The SUVs presented in iPlan are directly imported from the PET scanner and were compared between both to ensure precision. The PET activity was assessed semiquantitatively to assess the PET activity at a given focus. The implementation of the SUV calculation in the PET scans in the iPlan software (version 3.0) was based on [18]. iPlan displayed the SUV information per pixel as calculated and exported by the scanner. The displayed values were compared with the SUV obtained directly at the scanner before use.

Uptake in the contralateral brain (VOI), thalamus and choroid plexus (mean and maximum), arteries, and veins was also defined using Syngovia (Siemens Healthineers, MI Neurology Workflow, Siemens Healthcare GmbH, Erlangen, Germany) and compared with the values collected in iPlan. Figure 1B shows the VOI in the contralateral brain defined using Syngovia in early and late PET acquisitions. The regions of interest (ROIs or VOIs) were defined as follows and as in Figure 1C: brain, VOI sphere (10–20 cm^3^) in the contralateral brain; middle cerebral artery, measured as the maximal point uptake value in the ROI circle (3–5 mm^3^) based on T2_-weighted turbo spin-echo or gadolinium-enhanced T1-weighted turbo spin-echo (T1_CE or T2) in the contralateral brain; and sinus, measured as the maximal point value in the sagittal and sigmoidal sinuses in the VOI sphere (0.3–0.8 cm^3^) in the largest visible uptake area based on a visual assessment on PET based on T1_CE or T2. The thalamus was measured as the mean SUV and max SUV found in the VOI sphere (0.5–1 cm^3^) at the area of the highest area of uptake defined on a visual assessment based on T1_CE or T2 in the contralateral brain. Other basal ganglia were checked in random patients, but the values were similar. The choroid plexus was defined as the max values in the 0.3–0.6 cm^3^ ROI circle. The resolution (spatial resolution) of the Biograph mMR scanner declared by the scanner manufacturer is 4.6 mm in the central field of view (according to the NEMA standard for FBP, 344 × 344 matrix). In our imaging, a 344 × 344 matrix was used—to improve the resolution an iterative algorithm was additionally used—taking into account the angle of incidence of the LOR on the detector block (HD PET and OSEM 3D + PSF) and the appropriate statistics of counts were provided (acquisition duration 10 min). All the corrections affecting the quality of imaging were also applied in the reconstruction. It can be assumed that the resolution in this study was less than 4 mm. All the settings of the acquisition protocol were aimed at improving the resolution and reducing the occurrence of undesirable phenomena, such as partial volume effects.

Examples of biopsy trajectories and SUVs in different reference areas (background) are presented in Figure 1. At the second acquisition timepoint, all the VOI spheres/ROI circles were repeated by synchronizing the images and aligning the areas in the early and late acquisitions. The SUVs and TBR values of the samples taken from each trajectory (e.g., FLAIR area) in relation to different backgrounds were analyzed.

### 2.4. Classification Tree for Differential Diagnoses

Due to missing data, the analysis was ultimately performed on 262 cases. The tree was built using the C&RT method [19]. The FACT-type direct stop was adopted as the stop rule, and the Gini measure was chosen to assess the goodness of fit [20]. A *p*-value was computed for each end node for class membership (astrogliosis vs. glioma). The predictor with the smallest *p*-value was selected for partitioning its corresponding node. The global costs of cross-validation were lower than the costs of cross-validation for the selected tree and, likewise, the standard error of the global cross-validation costs was close to the standard error of the costs of the selected tree, proving that the selected tree was the correct size and not over-trained. In addition, the cost of resubstitution was similar to the cost of a cross-validation for the selected tree, confirming the selection of the best tree. The sensitivity, specificity, positive predictive value (PPV), and negative predictive value (NPV) were calculated.

### 2.5. Statistical Analysis

The normality of the distribution of variables was checked using the Shapiro–Wilk test. To compare the SUVs between structures, because the data were not normally distributed, the Friedman test with Dunn–Bonferroni multiple comparison tests were calculated. To compare the SUV between the grade or trajectory, the Kruskal–Wallis test with a Dunn–Bonferroni multiple comparison test were calculated. The Wilcoxon test was calculated to examine the differences between the two timepoints because the data were not normally distributed.

A receiver operating characteristic (ROC) analysis was performed for the optimal cut-off value for predicting glioma. The area under the ROC curve (AUC) with 95% confidence intervals (95% CI), sensitivity, specificity, PPV, and NPV were then calculated.

The calculations were performed using Statistica v13.3 (TIBCO, Palo Alto, CA, USA) and PQStat v.1.8.4.164 (PQStat Software, Poznan, Poland). A *p*-value < 0.05 was considered significant.

## 3. Results

The interval between PET imaging and tumor biopsy was a median of 9 days. In total, 284 samples were examined: 91 from T1-GAD, 110 from PET, 11 from PET^−^, and 72 from FLAIR areas. The final histopathology results based on all the targeted areas are presented in Table 1. Astrogliosis was significantly more common within the FLAIR regions, with over 50% of all the samples taken from FLAIR specific to astrogliosis. Therefore, we next sought to quantify the uptake that distinguished tumor from astrogliosis.

### 3.1. SUVs and TBR Values 10 and 60 min a.r.i. According to Biopsy Site

The SUVs and TBR values were significantly lower in the PET^−^ and FLAIR regions, but there were no significant differences between the T1-GAD and PET or PET^−^ and FLAIR regions, and this was not related to background reference measurements (Appendix A). Within the FLAIR and PET- areas, the SUV values were significantly lower, and in parallel, an astrogliosis diagnosis was more common, as shown in Table 2.

All the SUVs and TBR values at both timepoints (10 and 60 min) related to the trajectory are presented in Appendix A and Appendix A**.**


### 3.2. Differences in SUVs between Different Anatomical Structures at 10 and 60 min

All the SUVs measured in various anatomical structures are presented in Table 3 and Figure 1D. At both early (10 min) and standard (60 min) timepoints, the median SUVs of different structures varied but were significantly higher (*p* < 0.001) in the middle cerebral artery, thalamus, plexus, and sinus than in the contralateral brain. Furthermore, there were significant differences in uptake between most of these structures at both timepoints (Figure 1E).

### 3.3. Differences in SUV Related to Tumor Grade

The SUVs at both timepoints specific to tumor grade are presented in Table 4. There were significant differences in the SUV according to grade, but there was no difference between astrogliosis and grade 2 lesions (Figure 1E). All the figures and *p*-values related to the SUVs in tumors (including different grades), areas of astrogliosis, and in different anatomical structures are presented in the Appendix A. 

### 3.4. Target-to-Background Ratios

The TBRs of the anatomical structures were significantly different according to different tumor grades (Figure 2A). However, the TBRs had only limited value in differentiating grade 2 samples from astrogliosis, with only the TBR plexus (10 and 60 min a.r.i.) significantly different between the astrogliosis and grade 2 samples (*p* = 0.006 and *p* = 0.046, respectively). Detailed values of all the TBRs in relation to astrogliosis and tumor grade with specific *p*-values are presented in Appendix A.

### 3.5. Accuracy of Differentiating Tumor from Astrogliosis

An ROC analysis was performed to determine the optimal threshold for differentiating tumor from astrogliosis. An ROC analysis was performed for all the samples (including the samples taken from contrast-enhancing areas), PET^+^ and FLAIR (only samples outside contrast enhancement), and FLAIR (only samples outside PET hotspots).

### 3.6. Infiltration Defined by Standard (Single) Acquisition of FET-PET

A standard TBR using the contralateral brain as reference showed a high AUC (0.80; 95% CI 0.74–0.86 and *p* < 0.001). However, the routinely used threshold of 1.6 times the contralateral brain had a low specificity of 0.36 and high sensitivity of 0.91. Using early acquisition data with the same background as the reference and threshold produced similar results, with similar AUC of 0.81 (Table 5).

When the analysis was restricted to samples outside the contrast-enhancing and FET hotspots within FLAIR, the TBR brain accuracy significantly decreased to an AUC of 0.58 (Appendix A). Based on the TBR brain, the model could not differentiate astrogliosis from grade 2 tumor within FLAIR. 

### 3.7. Infiltration Defined Using Different Background Reference Structures

When considering different background structures, the highest AUC was related to the SUV10/SUV10 plexus ROI mean (AUC 0.87; 95% CI 0.82–0.92 and *p* < 0.001), with a sensitivity of 0.94 and specificity of 0.58 and a cut-off of > 1.0 (according to the Youden index) in early acquisitions (example in Figure 2C). Similarly, after 60 min, the highest AUC was with the SUV60/SUV60 plexus (0.82; 95% CI 0.77–0.88), with a sensitivity of 0.91 and specificity of 0.38 for a threshold value of >1.2 for the mean uptake in the contralateral plexus (Table 5).

Additionally, when the analysis was restricted to the samples collected only from the FLAIR trajectory, the TBR plexus ROI mean 10 min a.r.i. had the largest AUC (90% sensitivity and 61% specificity, cut-off > 1.0, AUC 0.75, 95% CI 0.63–0.87, and *p* < 0.001; Appendix A) for differentiating astrogliosis from any grade of tumor and specifically for differentiating astrogliosis from grade 2 tumor (100% sensitivity and 61% specificity, cut-off 1.03, and AUC 0.77). The same analysis performed for the standard TBR 1.6 resulted in 56% sensitivity and 41% specificity, AUC 0.58, and 95% CI 0.44–0.72 (Appendix A).

We also compared the values of the TBR plexus (maximum values) and TBR plexus ROI (mean values), which were significantly different (Appendix A), so we added the TBR plexus ROI to the ROC analysis presented below.

### 3.8. Comparison of Different Tumor-to-Background Ratios

A comparison of the ROC curves showed significantly better accuracy for the TBR plexus than the TBR brain in the early (*p* < 0.001) and standard acquisitions (*p* < 0.006) (Figure 2B). An ROC curve comparison of the TBR plexus in determining the glioma extent with respect to the acquisition time (PET 10 or PET 60) showed a trend to better accuracy in the early acquisitions (Appendix A). An example of tumor infiltration defined using the early TBR plexus is presented in Figure 2C. The accuracy of distinguishing tumor from astrogliosis based on different background structures was highest for the TBR plexus max and mean at both timepoints, as shown in Table 5.

### 3.9. Infiltration Defined by Dual Acquisition

Using dual acquisition there was a slight improvement in sensitivity and a decrease in specificity with respect to a single late acquisition. An example case comparing single and late acquisition with contrast enhancement and FLAIR hyperintense areas is presented on Figure 2D.

There were further improvements based on a dual TBR plexus threshold > 1.0 in early acquisitions and >1.2 in late acquisitions. The sensitivity and PPV were high, and the NPV increased from 0.54 to 0.67.

Comparison of tumor infiltration contours defined using contrast enhancement +2cm (as routinely used for radiotherapy planning), FLAIR hyperintense areas, TBR plexus ROI mean 10, standard TBR 1.6 and visual assessment is presented on Figure 2E.

### 3.10. Classification Trees to Differentiate Tumor and Astrogliosis at the Tumor Border

We next analyzed over 30 combinations of parameters from dual acquisitions to obtain the optimal accuracy for defining the glioma extent using classification trees. The optimal model is presented in Figure 2F. This model combines tumor-to-plexus ratios in standard acquisitions (SUV60/plexus60) with the absolute difference in the TBR plexus between both timepoints 10–60 a.r.i. This model achieved a 96% sensitivity, 61% specificity, 92% PPV, and 80% NPV. 

### 3.11. Differentiating Tumor Infiltration within Thalamus 

The thalamus is characterized by a naturally higher uptake than the brain cortex, as shown in Figure 3A. We next determined the optimal method to differentiate infiltration within the thalamus. The SUVs within the tumor were significantly higher than the mean SUVs in the contralateral thalamus (*p* < 0.001 in both acquisitions, Figure 3B), which resulted in a high accuracy when using absolute SUVs to differentiate the tumor from the thalamus, which rarely had an uptake above 1.1 (Figure 3C). The ROC analysis showed that the optimal threshold was ≥1.6 in standard acquisitions (AUC 0.93; Figure 3D).

### 3.12. Software Impact on Results

The TBR values generated by different software packages (iPlan and Syngovia) were compared for the plexus and thalamus. There were no significant differences between the different software packages and methods. A comparison of the thalamus max defined in iPLAN and the thalamus max defined in Syngovia is presented in Appendix A.

## 4. Discussion

This biopsy-controlled study showed that differentiating tumor from astrogliosis with PET is feasible. Grade 2 areas (frequently present in contrast-enhancing gliomas), especially within FLAIR, limit PET accuracy for defining the border between tumor and astrogliosis. We previously showed that FLAIR outside PET hotspots had low sensitivity and specificity (15% and 29%, respectively) for detecting tumor [15]. Here, we present a method to define the tumor extent within this area with high accuracy based on low-to-moderate uptake. Early acquisition and the tumor-to-plexus ratio significantly improved the accuracy in defining the tumor extent, regardless of grade, and the approach was software-independent. In developing new algorithms, we increased the accuracy of both acquisitions. Finally, it was possible to exclude the normal thalamus and decrease the risk of underdiagnosing tumor within this structure of naturally higher uptake. 

Common indications for PET imaging include the definition of the optimal biopsy site, prognostication, and delineation of the tumor extent for surgery and radiotherapy [21]. The optimal method for delineating glioma infiltration with PET has yet to be defined, as there are only limited reliable data from the tumor border. Dynamic or dual-timepoint PET acquisition may further improve the delineation of tumor infiltration, as low-grade tumor tends to be more avid in late acquisitions, whereas high-grade tumor tends to be identifiable in earlier acquisitions [22,23,24]. As high FET concentrations in peritumoral vessels tend to affect early acquisitions [12], there is still some debate about the optimal acquisition timepoint and reference structures for discriminating tumor from astrogliosis. Physiological uptake within reactive astrogliosis may additionally limit the accuracy of FET-PET for precisely defining the glioma extent [25]. To our best knowledge, this is the first paper defining an exact threshold to differentiate glioma and astrogliosis in dual-timepoint PET acquisition based on serial biopsies of diffuse contrast-enhancing gliomas taken from PET, T1-GAD, and FLAIR.

The main aim of this study was to differentiate astrogliosis and thalamus from tumor. From the clinical perspective, it is important that the precision of FET-PET for tumor estimation is limited by the grade 2 component of the tumor, because these areas behave in a similar manner to astrogliosis and provide the largest source of error. In our previous analysis [15], we found that most diffuse contrast-enhancing gliomas have a grade 2 component, necessitating the accurate differentiation of tumor from astrogliosis outside contrast enhancement and FET hotspots. The tumor-to-plexus ratio has been analyzed in a few studies, but here we show that this parameter is useful for optimally differentiating grade 2 tumor from astrogliosis. Tumor—including grade 2 areas—shows significantly higher tumor-to-plexus ratios than astrogliosis in early acquisitions. Based on the ROC analysis, the tumor-to-plexus ratio 10 min after injection had the highest accuracy for differentiating tumor from astrogliosis.

The current guidelines have determined cut-off values for semiquantitative PET analyses (tumor SUV compared to healthy brain SUV), which is especially useful for radiotherapy planning because it enables semi-automated tumor delineation based on threshold values for FET uptake, providing a baseline for other studies. The most commonly used threshold for glioma volume definition by PET is a 1.6 × VOI in the contralateral brain [26], although other centers have used a threshold of 1.8× background activity for delineation [27]. 

The 1.6 cut-off is based on a biopsy-controlled study of cerebral gliomas in which a lesion-to-brain ratio of 1.6 best separated tumoral from peritumoral tissue [26]. This landmark study used a static, single acquisition 20–40 min a.r.i., and a TBR of a 1.6 × VOI in the contralateral brain had 92% sensitivity and 81% specificity. Based on our data, the 1.6 threshold is suboptimal for differentiating glioma from astrogliosis outside contrast enhancement, and the further the hotspot, the lower the accuracy with very limited value inside FLAIR and outside the hotspot. Furthermore, Pauleit et al. [26] only studied a small number of samples from moderate uptake areas and targeted areas without FET uptake (mean TBR 1.1), whereas in our study the FLAIR-targeted areas showed moderately increased uptake (TBR 1.9–2.0). Consequently, Pauleit et al. [26] detected astrogliosis in 88% of samples taken from FLAIR areas, whereas we detected astrogliosis in 55% of the samples taken from the FLAIR areas. Moreover, Pauleit et al. [26] analyzed 52 samples from tumor areas and 26 from astrogliosis areas, compared with 230 and 54 here. Indeed, Pauleit et al. [26] only analyzed eight samples between 1.1 and 1.6, and, within this range, we found that most were astrogliosis or grade 2 tumor in standard acquisitions. On the other hand, the Pauleit study and many subsequent studies in the literature were based on a 20–40 min PET acquisition time frame (post-injection), and in our study, a later (40–60 min) time was used, without testing the 20–40 min time in between. Based on FET uptake kinetics, the peak occurs around 5–15 min post-injections and decreases slowly between 20 and 60 min. Moreover, uptake within grade 2 gliomas increases with time; therefore, later time frames should further increase thresholds between astrogliosis and grade 2. Therefore, it is rather unexpected that this may lower the accuracy of our later timepoint.

We found that the choroid plexus was the optimal anatomical reference (background) structure, as 10 min a.r.i the choroid plexus uptake was similar to astrogliosis, the maximum values in the contralateral brain, and significantly higher than the mean values in the brain. 

A recent prospective study analyzed the accuracy of static amino acid FET-PET and multiparametric MRI for detecting the glioma extent in relation to regional tissue samples using 174 biopsies from 20 patients with newly diagnosed gliomas. The authors showed that the infiltration zone of enhancing gliomas was best reflected by a combination of FET-PET and ADC mapping. Nevertheless, the poor result for FET-PET in delineating the glioma extent was unexpected and limited by the high percentage of low-grade gliomas (40%) in the study [12]. This may explain our finding that discrimination of grade 2 tumors at the infiltrating border zone is particularly difficult. In another study, newly diagnosed glioblastoma infiltration was greater in FET-PET than in rCBV maps (*p* < 0.001) or contrast enhancement with low spatial similarity of both imaging parameters [28], which is consistent with our experience. Hayes et al. [4] recently showed that in 83% of cases, FET-PET with the 1.6 threshold of symmetrical brain extends outside FLAIR volumes and in 71% outside clinical target volumes in grade 4 diffuse gliomas. Tumor-to-plexus-based volumes also significantly differed from 1.6 threshold-based volumes and provided more logical images of the infiltration. The infiltration extent may penetrate outside routine CTV (enhancement +2 cm) for glioblastoma (Figure 2E).

A post-mortem study compared whole-brain histology with in vivo dynamic FET-PET and MRI in a patient with glioblastoma. FET-PET reliably identified viable tumor tissue in all parts of this large, progressive lesion and allowed for the differentiation of areas with pronounced astrogliosis and only minor, scattered neoplastic cells using the 1.6 threshold [29]. Similar to our findings, astrogliosis showed a higher SUV typical to grade 2 gliomas.

Grade 2 and 3 diffuse gliomas are frequently non-enhancing, and the definition of the tumor extent relies on signal abnormalities in T2-weighted MRI or in the FLAIR sequence. It is well known, however, that neither contrast enhancement nor FLAIR abnormalities are sensitive for neoplastic tissue. Pauleit et al. [26] showed that contrast enhancement had a 38% sensitivity and 96% specificity and FLAIR a 96% sensitivity but only a 4% specificity. In our recent study [15], the sensitivity and specificity of the contrast enhancement and FLAIR were 38% and 93% and 15% and 30% (outside FET hotspots). Many studies have demonstrated that a considerable amount of tumor may extend beyond areas of contrast enhancement and that FLAIR abnormalities may exceed the tumor extension [30]. We showed that almost 40% of our contrast-enhancing gliomas had a grade 2 component, creating a serious limitation in terms of the optimal strategy to delineate these diffuse gliomas. While FLAIR may be used, we previously reported that PET-avid areas can be found outside FLAIR. Here, we found that grade 2 and astrogliosis have similar uptake values, and a model that included uptake kinetics inside an area of interest, plexus, and astrogliosis increased the accuracy for differentiating this area. From the clinical perspective, astrogliosis is much smaller than FLAIR, so a slightly lower specificity with high sensitivity for finding all tumor cells is feasible and acceptable.

In the dual-timepoint analysis, our data showed that uptake in astrogliosis increased over time, similar to uptake in grade 2 tumor, brain, and thalamus. While we applied data-mining methods to try to improve the diagnostic accuracy by adding secondary acquisition data, for most single acquisition analyses—even for grade 2 components—early acquisition achieved better results than late acquisition.

The plexus behavior was more similar to grade 3 and 4 tumors and slightly decreased over time, so it was the most accurate reference structure to discriminate glioma (regardless of grade) from astrogliosis. Especially within FLAIR areas, this was a very accurate method to define the tumor extent and relatively simple to incorporate in practice (2D ROI). However, as we showed in Table 5, the overall accuracy was the highest when the plexus was used as the control value, and the actual specificity values remained quite low (0.58–0.65); this means that quite large areas outside the tumor region could be identified (falsely) as being tumorous. This is acceptable if the goal is not to miss tumor tissue at the price of overestimating the tumor; still, the latter can pose challenges in areas where targeted treatment is close to eloquent areas and the removal or damage of these non-tumorous regions would cause postoperative deficits. Nevertheless, the accuracy of the TBR plexus significantly improved the FLAIR accuracy and should replace the FLAIRectomies strategies that are of increasing interest recently. Also, other imaging modalities such as ADC in combination with FET-PET were shown to be beneficial and may improve diagnostic accuracy outside CE [31]. 

Some normal anatomic brain structures such as the thalamus may exhibit higher FET-PET uptake than the parietal lobe serving as a background reference tissue [32]. Our SUV analysis confirmed that observation. Penetration into or close to the thalamus may limit the accuracy of the method and defining infiltration in this area based on tumor-to-plexus ratios. However, uptake higher than 1.6 times the contralateral thalamus in standard acquisitions should be considered pathological with high accuracy. 

This study had some limitations. There may be concerns regarding the accuracy of extracting activity within the small lesions, e.g., middle cerebral arteries. The resolution may indeed approach the borderline for such tasks. In our study, we did not employ activity correction based on lesion diameter using vessel segmentation from structural MRI. This could be considered a limitation of our methodology, particularly in the context of accurately capturing activity within middle cerebral arteries. It is important to note that the SUV is dependent on the spatial resolution of the PET scanner and image processing; however, it is more important to define maximum SUVs, so mostly mean values were analyzed here. However, absolute values are not comparable between different centers and must be considered with caution [12], and the threshold-based method as a relative value should overcome this limitation [33]. Although, the specificity and sensitivity may vary according to the software used due to minor differences. For the SUV, the overall outcome was similar. Also, some medications (e.g., dexamethasone) may influence FET uptake [34], which must be considered in future analyses. Overall, the advantage of our study is the transferability of the findings into clinical practice as threshold FET uptake values are provided to facilitate the interpretation of individual patient data regardless of the software used.

## 5. Conclusions

The low accuracy of MRI in determining glioma extent negatively impacts the efficacy and safety of the current therapies. We show that improvements are possible with dedicated PET protocols. We present a method to accurately define the infiltration of glioma based on FET-PET and referring to uptake in the choroid plexus. Specifically, the tumor-to-plexus approach applied to the early acquisition of FET-PET improves the definition of glioma extent for surgery or radiotherapy. This method of gross tumor volume definition increases accuracy compared with T1-GAD or FLAIR-based volumes and was confirmed pathologically. This study provides a background to further explore the efficacy and safety of such personalized volumes in various local therapies. The tumor-to-plexus approach is relatively easy and reproducible for implementation in planning protocols (Figure 3E).

## Figures and Tables

**Figure 1 cancers-16-01265-f001:**
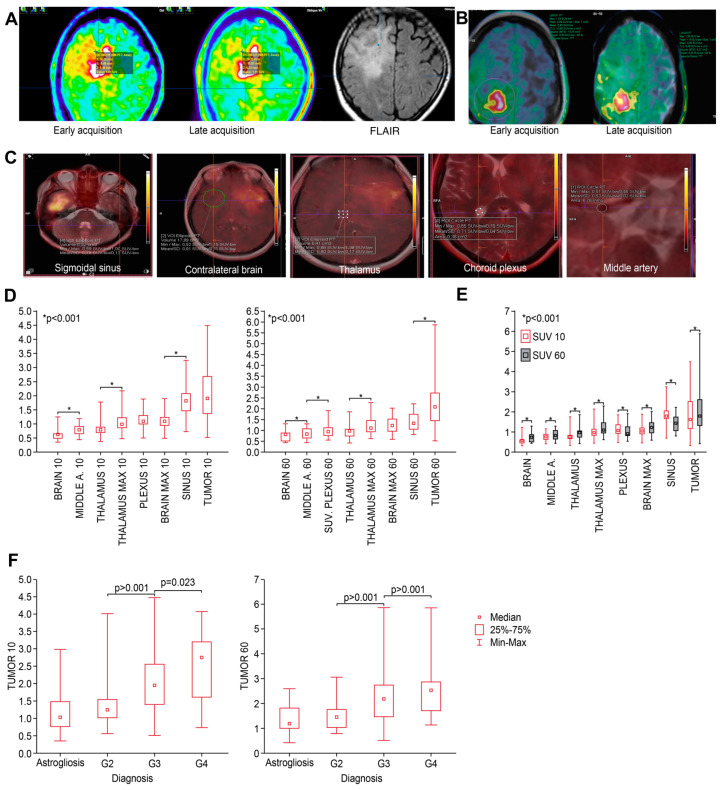
(**A**) An image from biopsy planning system—a representative case of biopsy targeting (blue dot) different SUVs at the tumor border in early (**left**) and late (**middle**) PET and FLAIR MRI (**right**) sequences. (**B**) Background VOI parameters (contralateral brain) defined at both timepoints. (**C**) Definition of different reference as indicated. (**D**) Median SUVs of different structures measured at both timepoints (early—left, standard—right); structures are presented from lowest to highest values; and *p*-values indicate the closest differences that are significant. (**E**) Comparison of SUVs in different structures between early and late timepoints. SUVs were significantly higher at T60 than at T10. (**F**) At both timepoints, median SUVs significantly discriminated between grades of tumor but not astrogliosis.

**Figure 2 cancers-16-01265-f002:**
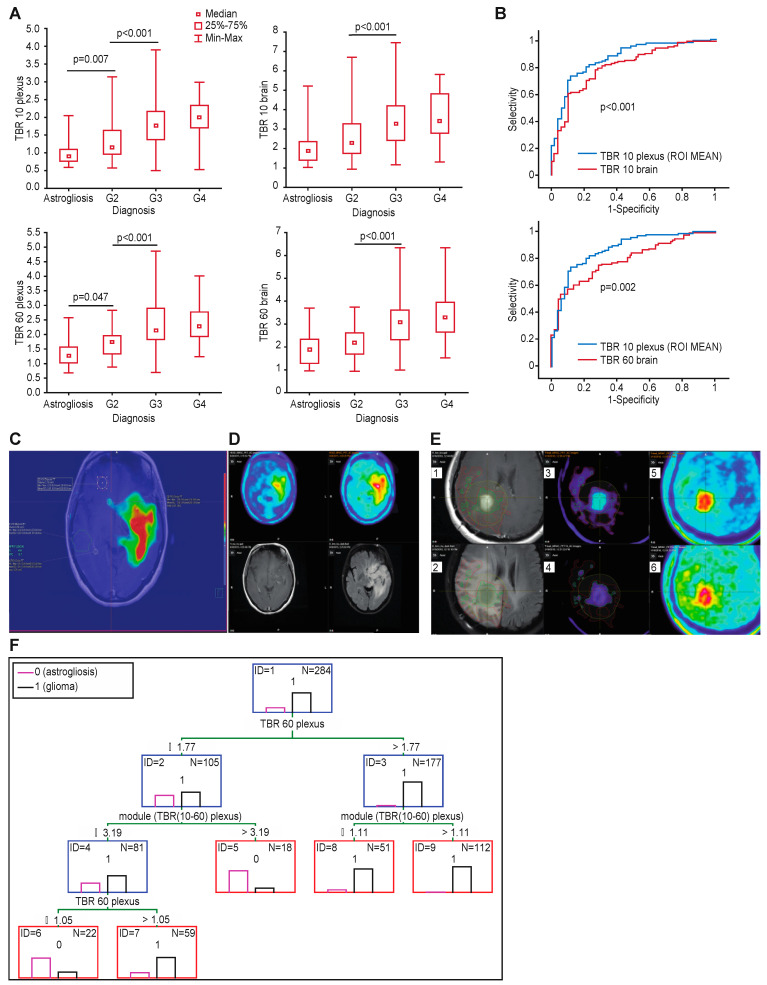
(**A**) TBRs of the various anatomical structures were significantly different according to different tumor grades. (**B**) Comparison of ROC curves for TBR plexus and TBR brain. TBR plexus shows higher accuracy in all cases. (**C**) Glioma infiltration extent defined based on plexus uptake >1 in early image shows that infiltration penetrates the brainstem, consistent with the FLAIR image, but also penetrates into the other hemisphere through the anterior commissure. (**D**) Comparison of early and late acquisition of FET PET (upper row) and T1-CE and FLAIR (lower row)—visual assessment (**E**) Contours comparison—(1) Delineation of glioma extent based on contrast enhancement + 2 cm margin (deep yellow contour), upper left image; (2) FLAIR hyperintense area related to contours defined using CE+2cm, TBR plexus ROI mean 10 and standard TBR 1.6, lower left image; (3) tumor contour based on TBR plexus ROI mean 10 ratio > 1× mean uptake within plexus on early FET-PET image (red contour), upper middle image; (4) contour based on standard TBR 1.6× contralateral brain (light green contour), lower middle image; (5) unadjusted early and (6) standard FET PET images, upper and lower right images . (**F**) Optimal classification tree to differentiate tumor and astrogliosis at the tumor border. TBR plexus > 1.77 and module > 1.11 are characterized for tumor samples, and TBR plexus 60 between 1.05 and 1.77 and module < 3.19 are specific for tumor.

**Figure 3 cancers-16-01265-f003:**
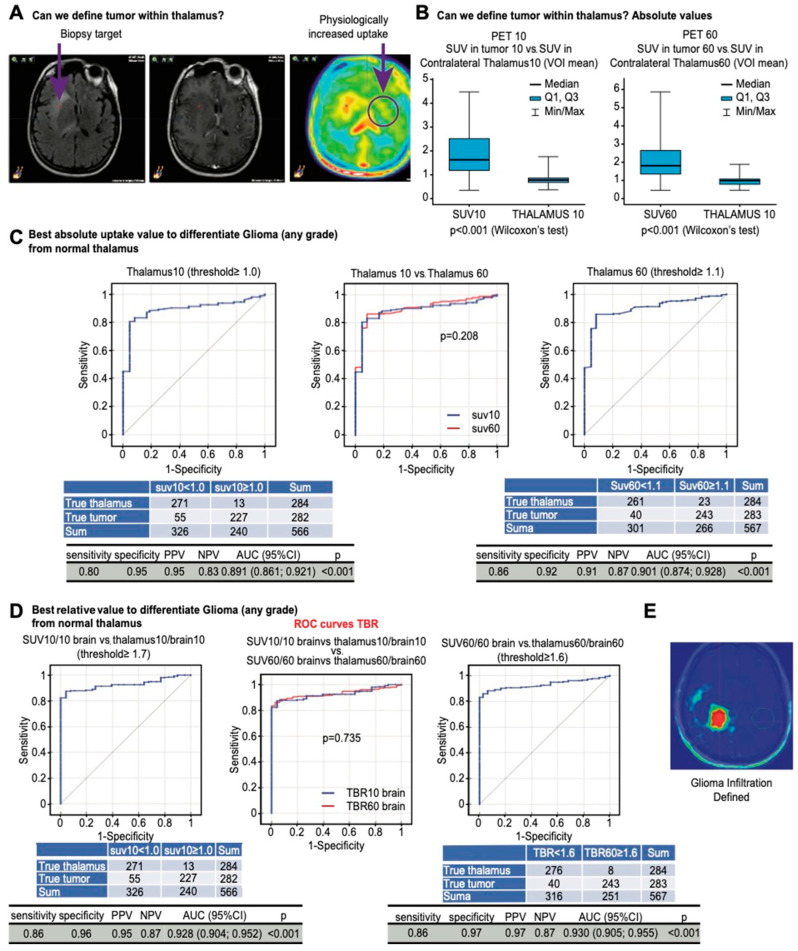
(**A**) Correlations between SUVs in tumor samples, astrogliosis, and uptake in the contralateral basal ganglia (circle) and thalamus presenting with physiologically increased uptake. (**B**) Contralateral thalamus has significantly lower uptake than tumor samples at both timepoints. (**C**) ROC curves showing the accuracy in differentiating tumor samples from normal thalamus with an absolute uptake value > 1 and >1.1 in early and late acquisitions. (**D**) ROC curves showing the accuracy in differentiating tumor samples from normal thalamus with relative uptake values ≥ 1.7 and ≥1.6 in early and late acquisitions. (**E**) An image of glioma infiltration defined using early TBR plexus.

**Table 1 cancers-16-01265-t001:** Final histopathological diagnoses and number of samples taken from each area.

Patient	Number of Samples Analyzed from Each Trajectory				Final Histopathology Result (WHO 2021)
	T1-Gad+	PET+	PET-	FLAIR+	
1	7	8	0	8	Oligodendroglioma, IDH-mutant, G3
2	8	8	0	7	Astrocytoma, IDH-mutant, G3
3	7	8	6	3	Oligodendroglioma, IDH-mutant, G3
4	5	4	0	4	Astrocytoma IDH-mutant, G3
5	4	4	0	6	Glioblastoma, IDH wildtype, G4
6	3	3	0	2	Glioblastoma, NOS, G4
7	3	3	0	5	Astrocytoma, IDH-mutant, G3
8	4	4	0	7	Glioblastoma, NOS, G4
9	4	5	0	5	Oligodendroglioma, IDH-mutant, G3
10	3	3	5	5	Glioblastoma, IDH wildtype, G4
11	3	6	0	8	Astrocytoma, NOS, G2
12	4	4	0	3	Oligodendroglioma, IDH-mutant, G3
13	4	4	0	0	Glioblastoma, IDH wildtype, G4
14	3	4	0	4	Astrocytoma, IDH-mutant, G3
15	6	6	0	4	Astrocytoma, IDH-mutant, G4
16	0	7	0	3	Astrocytoma, IDH wildtype, G2
17	7	3	0	2	Astrocytoma, IDH-mutant, G4
18	4	4	0	3	Glioblastoma IDH wildtype G4
19	5	4	0	4	Glioblastoma IDH wildtype G4
20	6	5	0	5	Oligodendroglioma, IDH-mutant, G3
21	0	5	0	1	Oligodendroglioma, IDH-mutant, G3
22	3	1	0	2	Glioblastoma NOS, G4
23	0	7	0	3	Astrocytoma, IDH-mutant, G4

**Table 2 cancers-16-01265-t002:** Number of samples with each histopathological diagnosis and trajectory-specific mean SUVs and TBR values 10 and 60 min a.r.i.

HP	Overall		T1-GAD		PET		PET-		FLAIR	
	N	%	N	%	N	%	N	%	N	%
Astrogliosis	54	19	4	4	9	8	3	27	38	53
G2	45	16	5	6	21	19	6	55	13	18
G3	125	44	55	60	51	46	2	18	17	24
G4	60	21	27	30	29	27	0	0	4	5

**Table 3 cancers-16-01265-t003:** Standarized uptake values at both timepoints in various anatomical structures.

SUV	N	Mean	Median	Min	Max	Lower Q	Upper Q	SD
PLEXUS 10	23	1.12	1.07	0.48	1.86	0.91	1.28	0.31
THALAMUS MAX 10	23	1.04	0.97	0.46	2.15	0.80	1.19	0.37
THALAMUS	23	0.77	0.77	0.35	1.76	0.60	0.86	0.27
BRAIN 10	23	0.60	0.60	0.33	1.23	0.47	0.69	0.20
BRAIN 10 MAX	23	1.09	1.05	0.47	1.88	0.92	1.22	0.32
Middle A. 10	23	0.78	0.77	0.42	1.17	0.64	0.90	0.19
SINUS 10	23	1.88	1.92	0.71	3.25	1.45	2.08	0.55
SINUS 60	23	1.46	1.45	0.82	2.23	1.07	1.83	0.41
PLEXUS 60	23	0.98	0.91	0.56	1.92	0.76	1.13	0.31
THALAMUS Max 60	22	1.17	1.10	0.63	2.30	0.90	1.39	0.40
THALAMUS 60	23	0.90	0.88	0.43	1.86	0.72	1.05	0.30
BRAIN 60	23	0.73	0.74	0.44	1.31	0.51	0.88	0.23
BRAIN 60 MAX	23	1.20	1.17	0.60	2.03	0.83	1.48	0.37
MIDDLE A. 60	23	0.85	0.82	0.45	1.31	0.60	1.08	0.26

**Table 4 cancers-16-01265-t004:** SUVs in individual samples specific to tumor grade at both timepoints.

HP	Timepoint	N	Mean	Median	Min	Max	Lower Q	Upper Q	SD
Astrogliosis	SUV10	52	1.12	1.04	0.34	2.98	0.76	1.47	0.54
	SUV60	53	1.40	1.21	0.43	2.61	1.02	1.83	0.57
G2	SUV10	45	1.35	1.23	0.55	4.02	1.01	1.54	0.60
	SUV60	45	1.60	1.47	0.79	3.07	1.05	1.77	0.62
G3	SUV10	125	2.05	1.96	0.50	4.47	1.39	2.55	0.89
	SUV60	125	2.27	2.18	0.53	5.86	1.47	2.75	0.99
G4	SUV10	60	2.50	2.75	0.72	4.08	1.60	3.20	0.87
	SUV60	60	2.60	2.56	1.15	5.86	1.72	2.87	1.17

**Table 5 cancers-16-01265-t005:** Accuracy of distinguishing tumor from astrogliosis based on different background structures.

Timepoint	TBR	Cut-off	Sensitivity	Specificity	PPV	NPV	AUC (95% CI)	*p*-Value
10 min	TBR	1.60	0.93	0.37	0.87	0.56	0.809 (0.744; 0.874)	<0.001
	TBR max	1.60	0.57	0.87	0.95	0.31	0.796 (0.730; 0.862)	<0.001
	TBR plexus	1.00	0.87	0.65	0.92	0.53	0.868 (0.820; 0.917)	<0.001
	TBR plexus ROI mean	1.00	0.95	0.58	0.90	0.71	0.869 (0.814; 0.923)	<0.001
	TBR thalamus	1.55	0.83	0.69	0.92	0.48	0.801 (0.735; 0.868)	<0.001
	TBR thalamus max	1.40	0.74	0.69	0.91	0.38	0.742 (0.668; 0.817)	<0.001
	TBR MIDDLE A.	1.90	0.71	0.85	0.95	0.40	0.831 (0.774; 0.889)	<0.001
	TBR sinus	0.74	0.77	0.81	0.95	0.44	0.843 (0.788; 0.899)	<0.001
60 min	TBR	1.60	0.91	0.36	0.86	0.48	0.797 (0.738; 0.856)	<0.001
	TBR max	1.60	0.57	0.75	0.91	0.29	0.7621 (0.698; 0.827)	<0.001
	TBR plexus	1.20	0.94	0.40	0.87	0.60	0.845 (0.790; 0.900)	<0.001
	TBR plexus ROI mean	1.20	0.91	0.38	0.86	0.51	0.828 (0.773; 0.883)	<0.001
	TBR thalamus	1.80	0.77	0.70	0.92	0.41	0.789 (0.725; 0.852)	<0.001
	TBR thalamus max	1.10	0.88	0.51	0.88	0.50	0.752 (0.680; 0.823)	<0.001
	TBR MIDDLE A.	2.10	0.64	0.79	0.93	0.34	0.788 (0.727; 0.849)	<0.001
	TBR sinus	1.30	0.63	0.89	0.960	0.35	0.793 (0.735; 0.850)	<0.001

## Data Availability

The data that support the findings of this study are available on request from the authors based on reasonable request [I.M., M.H.]. The data are not publicly available due to restrictions, e.g., their containing information that could compromise the privacy of research participants.

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
