# Peer review of "A Biopsy-Controlled Prospective Study of Contrast-Enhancing Diffuse Glioma Infiltration Based on FET-PET and FLAIR"

_cancers, 2024, doi:10.3390/cancers16071265_

Round 1
Reviewer 1 Report
Comments and Suggestions for Authors
In this paper, Harat et al. present a detailed analysis to optimize the use of dual-acquisition FET-PET to define gliomas, specifically, to differentiate histopathologically defined glioma tissue from gliosis, and also to define tumor in the thalamus (that has physiologically high FET uptake). While conventional FET-PET analysis commonly uses contralateral brain as the control tissue (to create tumor-to-brain, TBR) values, the current study found that the use of contralateral plexus may improve the differentiating accuracy of FET PET. The main strength of the study is the availability of a large number of image-guided biopsy samples from 23 patients (the same data set that was utilized in a previously published study of the group); and also the use of two PET acquisition time points combined with a variety of control tissues/structures to attempt to find the optimal variable to achieve the differentiation set out in the aims. Overall, the use of plexus activity has led to a moderate improvement, although the obtained specificity values suboptimal, while sensitivity exceeded 0.90. The methods and results are dessribed in great details, and the findings provide some novel aspects that may improve the clinical utility of FET PET in glioma imaging; but some aspects should be refined or explained more clearly, as detailed in the specific comments below.
Specific comments:
The title suggests that this is a study on high-grade gliomas. Still, two of the 23 patients (#11 and 16 in table 1) had a grade 2 (i.e., low-grade) glioma (astrocytoma) as the final histopathology diagnosis. Please reconcile this.
Abstract: Please add the number of patients the samples were obtained from.
Also: the abstract states that “… differentiated from grade 2 glioma with 90% sensitivity and 61% specificity, with dual acquisition improving the sensitivity to 94%.” – Here it would be important to also add the specificity from dual acquisition, to provide a more complete picture for the readers.
Introduction: It would be helpful to define the study aims more specifically. Indeed, the main aim of the study “was to differentiate astrogliosis and thalamus from tumor” tissue, as stated in the Discussion. To achieve this, the authors used not only dual time point acquisitions but also tested a variety of control tissues to obtain optimized thresholds. This specific information would be useful to be included already in the Introduction, so the readers understand why the various contralateral structures were analyzed, when this comes up first in the Methods.
Methods:
Acquisition timing: The authors discuss in the Discussion that the original FET PET study (ref. 26), that defined the 1.6 cutoff TBR threshold used in many subsequent studies in the literature, was based on a 20-40 min PET acquisition time frame (post-injection). The current study did not find this TBR (1.6) threshold accurate to differentiate tumor from non-tumor tissue. However, different time frames were used in the present study (although their use is not clearly explained): a much earlier (5-10 min) and a later (40-60 min) time, without testing the 20-40 min time in between. Therefore, it remains unclear how much of the lower accuracy was due to this timing difference.
Also, in their previous study of the authors on the same cohort of patients (ref. 15), the authors reported the early acquisition to be completed at 5-15 min. Please reconcile this.
Definition of biopsy trajectories: on page 4, lines 120-122, the various MRI/PET combinations are described (trajectory 1-4); here the trajectories are defined, for PET, as being either PET+ or PET-. But it is not clearly specified here how PET+ vs. PET- was defined. PET- is said to have “low to moderate uptake”, as opposed to being a “hot spot” (PET+). But it is not clear what SUV (or other) threshold was used to call a biopsy spot PET- vs. PET+; or was it based on a subjective (visual) judgment? If the latter, how reliable/reproducible this judgment was? This needs to be clarified.
Definition of contralateral regions/structures: First, the authors should explain in the Methods why are they drawing these various contralateral structures. Or this could be set up in the Introduction (as mentioned above), so that the readers are informed that they were looking to optimize the control region – this is not clear at this point of the Methods. Then Fig. 1 B is said to show VOI in the contralateral brain (at two time points), but the images show no VOI and is also missing a color scale needed to appreciate the meaning of color differences (and the units rendered to the scale). On Fig. 1 C, the contours and definition of the “middle artery” is difficult to see.
Also, the authors should also not use the abbreviated names (T2_tse_tra, T1_tse_tra-Gad), instead, the full name of these sequences should be spelled out.
A main goal of the study was to optimize the differentiation between glioma vs. gliosis (based on histopathology). A common issue, e.g., on FLAIR images, is the inability to differentiate the tumor tissue from vasogenic edema that can have similar signal changes on FLAIR. How does the (non-tumorous) edema region classified by histopathology, does it always show gliosis, or may show other abnormalities that were not captured by histopathology?
Results: line 196: “Astrogliosis became more common with decreased uptake values” (Table 2). It is not clear what the authors meant to have “decreased uptake values” here. Table 2 shows number of samples with “PET” and “PET-“, and it is unclear how this is reflecting “decreased uptake values”. This sentence should be reworded to avoid confusion.
Table 1: What do the superscripts (2) represent in the table (in the column with final histopathology)?
Figure 2. C-D-E: PET scans again show color maps, but there is no color scale indicating what values the actual colors mean. Also, the legend for Figure 2C states that the scan shows “Glioma infiltration extent defined based on plexus uptake >1 and >1.2 in early and standard acquisitions; however, only a single scan is shown, rather than the two acquisitions implied in the legend. Please explain or modify this panel.
Table 5 shows the accuracy of the various variables to differentiate tumor from gliosis. While the overall accuracy was the highest when the plexus was used as the control value, the actual specificity values remained quite low (0.38-0.65); this means that quite large areas outside the tumor region could be identified (falsely) as being tumorous. This is acceptable if the goal is not to miss tumor tissue at the price of overestimating the tumor; still, the latter can pose challenges in areas where targeted treatment is close to eloquent areas and removal or damage of these non-tumorous regions would cause postoperative deficits. This issue would be worth discussing in more depth.
“Differentiating tumor infiltration within thalamus.” – On Fig. 3A, it is stated that the images show “naturally higher uptake than brain” in the thalamus. Thalamus is part of the brain, so this statement should be reworded. Also, the structure circled and pinpointed by an arrow on this figure is located outside the thalami (likely in or close to basal ganglia); the thalami on this figure indeed show physiological high uptake (red and orange), but this case is more of a problem of basal ganglia infiltration rather than thalamus…
Figure 3D: The misspelled word “realtive” should be corrected to “relative”; Fig. 3E is a panel showing “glioma infiltration defined”, but there is no explanation to it in the figure legend.
Discussion: “..some medications (e.g., dexamethasone” may influence FET uptake [12],…” – Ref. 12 (Langen et al., Neuro-Oncology 2020) is an editorial commentary. I could not find any data presented in it referring to steroid (or other drug) effects on FET uptake. Can you use a more proper reference here, to support this statement?
Comments on the Quality of English Language
Terminology used in the paper: If the “middle artery” was the “middle cerebral artery (MCA)”, this latter term should be used. Also, instead of “sinus”, it would be better to use the term “venous sinus” in the text.
Author Response
|
Response to Reviewer 1 Comments
|
||
|
1. Summary |
|
|
|
Thank you very much for taking the time to review this manuscript. Please find the detailed responses below and the corresponding revisions/corrections highlighted/in track changes in the re-submitted files.
|
||
|
3. Point-by-point response to Comments and Suggestions for Authors |
||
|
Comments 1: The title suggests that this is a study on high-grade gliomas. Still, two of the 23 patients (#11 and 16 in table 1) had a grade 2 (i.e., low-grade) glioma (astrocytoma) as the final histopathology diagnosis. Please reconcile this.
|
||
|
Response 1: Thank you for pointing this out. We agree with this comment. Therefore, we removed a term high-grade from the title and changed into contrast-enhancing diffuse gliomas as term more precisely describe our group [please see new title]” |
||
|
Comments 2: Abstract: Please add the number of patients the samples were obtained from. |
||
|
Response 2: Agree. We have added the number of patients to the abstract. [line 31]
Comments 3: Also: the abstract states that “… differentiated from grade 2 glioma with 90% sensitivity and 61% specificity, with dual acquisition improving the sensitivity to 94%.” – Here it would be important to also add the specificity from dual acquisition, to provide a more complete picture for the readers. |
||
|
Response 3: Thank you for this comment. We have, accordingly, modified the abstract results to make it more evident. Analyzing an optimal strategy for infiltration volume definition astrogliosis could be accurately differentiated from tumor samples using a choroid plexus as a background. Early acquisition improved the AUC in many cases, especially within FLAIR from 56% to 90% sensitivity and 41% to 61% specificity (standard TBR 1.6 vs early TBR plexus).[Line 39-46]
Comments 4: Introduction: It would be helpful to define the study aims more specifically. Indeed, the main aim of the study “was to differentiate astrogliosis and thalamus from tumor” tissue, as stated in the Discussion. To achieve this, the authors used not only dual time point acquisitions but also tested a variety of control tissues to obtain optimized thresholds. This specific information would be useful to be included already in the Introduction, so the readers understand why the various contralateral structures were analyzed, when this comes up first in the Methods. Response 4 :We have shaped our aim accordingly: Therefore, the primary aim of this study was to differentiate tumor tissue from non-tumoral based on PET/MR images and secondary aim was to optimize acquisition and analysis protocols for defining glioma infiltration outside FET hotspot and inside FLAIR. To achieve this aim, we used not only dual time point acquisitions but also tested an uptake in variety of control tissues to obtain optimized thresholds. [line 104-109]
Comments 5: Acquisition timing: The authors discuss in the Discussion that the original FET PET study (ref. 26), that defined the 1.6 cutoff TBR threshold used in many subsequent studies in the literature, was based on a 20-40 min PET acquisition time frame (post-injection). The current study did not find this TBR (1.6) threshold accurate to differentiate tumor from non-tumor tissue. However, different time frames were used in the present study (although their use is not clearly explained): a much earlier (5-10 min) and a later (40-60 min) time, without testing the 20-40 min time in between. Therefore, it remains unclear how much of the lower accuracy was due to this timing difference. Response 5: Thank you for pointing this out, we added this to discussion section. On the other hand, Pauleit study and many subsequent studies in the literature, was based on a 20-40 min PET acquisition time frame (post-injection) and in our study a later (40-60 min) time, without testing the 20-40 min time in between. Based on FET uptake kinetics the peak occurs around 5-15 minutes post-injections and decreases slowly between 20 to 60minutes [31]. Moreover, uptake within grade 2 gliomas increases with time therefore later time frames should further increases thresholds between normal brain and grade 2. Therefore, it is rather unexpected that this may lower the accuracy of our later time-point. [line 480-487]
Comments 6: Also, in their previous study of the authors on the same cohort of patients (ref. 15), the authors reported the early acquisition to be completed at 5-15 min. Please reconcile this.
Response 6: Thank you, I have corrected this important typo, we have indeed performed an early acquisition within 5-15 minutes after injection. [line 164]
Comments 7: Definition of biopsy trajectories: on page 4, lines 120-122, the various MRI/PET combinations are described (trajectory 1-4); here the trajectories are defined, for PET, as being either PET+ or PET-. But it is not clearly specified here how PET+ vs. PET- was defined. PET- is said to have “low to moderate uptake”, as opposed to being a “hot spot” (PET+). But it is not clear what SUV (or other) threshold was used to call a biopsy spot PET- vs. PET+; or was it based on a subjective (visual) judgment? If the latter, how reliable/reproducible this judgment was? This needs to be clarified. Response 7. Thank you, it is a very important point, we have analyzed it in details and presented in the supplement. It was a subjective evaluation done by study neurosurgeon. However, as you will see the SUV in PET- was 0.87 and 1.12(median SUV in all studied samples), whereas in PET +1.91 and 2.11 in early and late acquisition, accordingly. The differences in uptake were statistically significant in all selected areas beside FLAIR and PET-. We have moved this data from main manuscript into supplement due to limited space. Please see : Supplementary materials. Part 2. Detailed values of PET parameters between trajectories at both timepoints Comments 8:
Definition of contralateral regions/structures: First, the authors should explain in the Methods why are they drawing these various contralateral structures. Or this could be set up in the Introduction (as mentioned above), so that the readers are informed that they were looking to optimize the control region – this is not clear at this point of the Methods.
Response 8: The explanation why we have defined various region we have added in the introduction [Line 105-109].
Comments 9 Then Fig. 1 B is said to show VOI in the contralateral brain (at two time points), but the images show no VOI and is also missing a color scale needed to appreciate the meaning of color differences (and the units rendered to the scale).
Response 9: Thank you for pointing this out, as in an orginal figure it was possible to notice however we agree that in the text the color differences and text resolution were poor. Therefore, we have modified this figure and we believe that the resolution is better now.
Comments 10: On Fig. 1 C, the contours and definition of the “middle artery” is difficult to see.
Response 10: We have not contoured separately the middle cerebral artery but we would like to show in this figure how we analyze SUV within this anatomical structure which we believe is clear to identify on the trimmed and enalrged MRI scan that is added now. Comments 11: Also, the authors should also not use the abbreviated names (T2_tse_tra, T1_tse_tra-Gad), instead, the full name of these sequences should be spelled out. Response 11: Names are presented in full [line 198,202 and 205]
Comments 12: A main goal of the study was to optimize the differentiation between glioma vs. gliosis (based on histopathology). A common issue, e.g., on FLAIR images, is the inability to differentiate the tumor tissue from vasogenic edema that can have similar signal changes on FLAIR. How does the (non-tumorous) edema region classified by histopathology, does it always show gliosis, or may show other abnormalities that were not captured by histopathology? Response 12: Our histopathologist have found only gliosis within the samples.
Comments 13: Results: line 196: “Astrogliosis became more common with decreased uptake values” (Table 2). It is not clear what the authors meant to have “decreased uptake values” here. Table 2 shows number of samples with “PET” and “PET-“, and it is unclear how this is reflecting “decreased uptake values”. This sentence should be reworded to avoid confusion
Response 13: We have moved this into another paragraph and reworded. within FLAIR and PET- areas SUV values were significantly lower and in parallel an astrogliosis diagnosis was more common, as shown in grey in Table 2 [Line 290-291]
Comments 14 Table 1: What do the superscripts (2) represent in the table (in the column with final histopathology)? Response 14: Deleted, thank you. Comments 15 Figure 2. C-D-E: PET scans again show color maps, but there is no color scale indicating what values the actual colors mean. Response 15: In Figure 2C the light green colors means uptake above Plexus >0,91 . Unfortunately I am not able to provide a color bar with exact values. Comments 15 Also, the legend for Figure 2C states that the scan shows “Glioma infiltration extent defined based on plexus uptake >1 and >1.2 in early and standard acquisitions; however, only a single scan is shown, rather than the two acquisitions implied in the legend Response 15: Thank you for pointing this out. The Figure 2c shows early acquisition. Legend was changed accordingly. Comments 16 Table 5 shows the accuracy of the various variables to differentiate tumor from gliosis. While the overall accuracy was the highest when the plexus was used as the control value, the actual specificity values remained quite low (0.38-0.65); this means that quite large areas outside the tumor region could be identified (falsely) as being tumorous. This is acceptable if the goal is not to miss tumor tissue at the price of overestimating the tumor; still, the latter can pose challenges in areas where targeted treatment is close to eloquent areas and removal or damage of these non-tumorous regions would cause postoperative deficits. This issue would be worth discussing in more depth. Response 16: We have added this point into discussion and contrast this results with FLAIRectomies that are gaining popularity. However, as we showed in Table 5 the overall accuracy was the highest when the plexus was used as the control value, the actual specificity values remained quite low (0.58-0.65); this means that quite large areas outside the tumor region could be identified (falsely) as being tumorous. This is acceptable if the goal is not to miss tumor tissue at the price of overestimating the tumor; still, the latter can pose challenges in areas where targeted treatment is close to eloquent areas and removal or damage of these non-tumorous regions would cause postoperative deficits. Nevertheless, the accuracy of TBRplexus improved significantly the FLAIR accuracy and should replace FLAIRectomies strategies that are of increasing interest recently. [Lines 540-550] Comments 17“Differentiating tumor infiltration within thalamus.” – On Fig. 3A, it is stated that the images show “naturally higher uptake than brain” in the thalamus. Thalamus is part of the brain, so this statement should be reworded. Response 17: The phrase is reworded. Now it is a brain cortex. [Line 396] Comments 17: Also, the structure circled and pinpointed by an arrow on this figure is located outside the thalami (likely in or close to basal ganglia); the thalami on this figure indeed show physiological high uptake (red and orange), but this case is more of a problem of basal ganglia infiltration rather than thalamus… Response 17: Exactly, to be precise the Figure 3A is a case of physiologically increased uptake within contralateral brain to biopsy site which in fact represents basal ganglia here. We have added to the legend that in circle there is increases uptake in basal ganglia, nevertheless the thalami uptake also is presented. In the study we focused on thalamus as showed in Figure 1C. Figure 3. (A) Correlations between SUVs in tumor samples, astrogliosis, and uptake in the contra-lateral basal ganglia (circle) and thalamus presenting with physiologically increased uptake. [line 406-407] Comments 18Figure 3D: The misspelled word “realtive” should be corrected to “relative”; Fig. 3E is a panel showing “glioma infiltration defined”, but there is no explanation to it in the figure legend. Response 18: Corrected, thank you, Figure 3 (E) An image of glioma infiltration defined using early TBR plexus. [Line 412]
Comments 19Discussion: “..some medications (e.g., dexamethasone” may influence FET uptake [12],…” – Ref. 12 (Langen et al., Neuro-Oncology 2020) is an editorial commentary. I could not find any data presented in it referring to steroid (or other drug) effects on FET uptake. Can you use a more proper reference here, to support this statement? Response 19: We have selected a more appropriate paper ref. 32:
Stegmayr, C.; Stoffels, G.; Kops, E. R.; Lohmann, P.; Galldiks, N.; Shah, N. J.; Neumaier, B.; Langen, K. J. Influence of Dexamethasone on O-(2-[18F]-Fluoroethyl)-L-Tyrosine Uptake in the Human Brain and Quantification of Tumor Uptake. Molecular imaging and biology, 2019, 21, 168–174. [Line 666]
|
||
Reviewer 2 Report
Comments and Suggestions for Authors
This is an interesting biopsy-controlled study conducted on an adequate sample of patients.
The authors should address the two following points:
#1 Table 1. Samples taken from PET negative areas belongs to two patients only. This is, in my opinion, a major limitation of the study which has not been sufficiently addressed in the paper.
#2 The authors found different uptakes by the anatomical structures in patients with different grades of disease. What is their explanation for that ? This has not apparently been discussed in the paper.
Minor remarks
# Page 13 Line 398. The sentence is not clear. Maybe “extends” instead of “extent” ?
Author Response
|
Response to Reviewer 2 Comments
|
||
|
1. Summary |
|
|
|
Thank you very much for taking the time to review this manuscript. Please find the detailed responses below and the corresponding revisions/corrections highlighted/in track changes in the re-submitted files. |
||
|
2. Point-by-point response to Comments and Suggestions for Authors |
|
|
|
Comments 1: Table 1. Samples taken from PET negative areas belongs to two patients only. This is, in my opinion, a major limitation of the study which has not been sufficiently addressed in the paper.
|
||
|
Response 1: “PET negative” areas were called and selected as biopsy target when contrast enhancement was present but no FET increased uptake was found inside. This scenario is very uncommon, that’s why, in only two patients we were able to find such biopsy target. Nevertheless, in our group we have biopsied and analyzed a plenty of samples (71 samples) with low to moderate PET uptake within FLAIR trajectory (please see the supplementary part 2) even lower than uptake within normal brain. Therefore, I believe it is something that you would like to see and this trajectory is a real strength of our study. Probably the nomenclature “PET negative” and FLAIR was misleading as both represents low to moderate uptake. |
||
|
Comments 2: The authors found different uptakes by the anatomical structures in patients with different grades of disease. What is their explanation for that ? This has not apparently been discussed in the paper. |
||
|
Response 2: We have found a different Target-to-reference (eg. Brain or choroid plexus) in relation to tumor grade. This is due to higher uptake within samples of higher grade of the disease. However we have not found difference inside anatomical structures in relation to final diagnosis.
Comments 3: Page 13 Line 398. The sentence is not clear. Maybe “extends” instead of “extent” ? Response 3: Corrected to extends, thank you [Line 503].
|
||
Reviewer 3 Report
Comments and Suggestions for Authors
The manuscript aims to show the added value of FET-PET in distinguishing tumor and non-tumor tissue in 23 patients with contrast-enhancing gliomas. The utility of CE-T1w, FLAIR, and FET-PET scan is shown and verified by using histology samples from the tumor and peri-tumor areas. The article also proposes new thresholds for evaluation, evaluates different methods for estimating reference background signal.
Overall, the paper is well written, however, there is a lot of tables, figures and results, which makes it difficult to read and shortening the results and discussion would be beneficial. The abstract can be improved to deliver a clear and easy-to-read message to the reader. There are several things that need to be addressed - please see further minor and major comments below.
ABSTRACT:
1. Please provide basic demographics - number of patients, age+sex, which grades were included?
2. Can you please explain what exactly this means? “increased sensitivity within FLAIR to”
3. The results should be more focused and clear. Perhaps include less than more. 1), “although uptake in astrogliosis and grade 2 glioma was similar.” similar, but still allowing differentiation? 2) “to 71% but specificity remained low” - how low? 3) “Defining an optimal strategy for defining infiltration volume based on the choroid plexus, “ - unclear how is this related to the stated goals. Or is the choroid plexus used as background?
METHODS/RESULTS
4. (MAJOR) Figure 1A - please adjust the image crop so that corresponding images cover a similar area. Also, make sure that any potentially sensitive information is removed - 1B is not really readable, but it is close.
5. Figure 1D - Unclear difference between the two panels - explain that it’s the two different time points. The caption says that a median SUV is provided; why are there “THAMALUS” and “THALAMUS MAX”.?
6. Line 109 - What was the brand of the scanner and MR field strength? TOF PET?
7. Line 137-141 - Relevance of this paragraph is not entirely clear to me. Was this used for biopsy planning, then it can be perhaps shortened and moved way up.
8. (MAJOR) Line 142 - What was the resolution of PET. In general, the resolution of PET is way too low to be used to check activity in intracranial arteries or even internal carotid arteries without facing partial volume effects, which complicates defining a reliable reference value. How was this dealt with? This might need to be listed as a limitation of this method. This also applies to samples from the tumor border - they all have lower SUV, but it is unclear if this is due to lower intake or simply because of partial volume effects on the thin peri-tumoral region.
9.Lines 163-178 - The paragraph needs more editing to make it clearer and easier to read.
10. (MAJOR) Table 1 - Unclear what classification was really used when IDH-mutant glioblastomas are reported. Also, was 1p/19q used to classify oligodendrogliomas?
11. (MAJOR) Figure 2 - The decision tree with repeated use of the same parameter might be overtrained and spliting training and validation dataset is a must. Panels G and H are missing.
DISCUSSION:
12. (MAJOR) Line 325 - unclear why the authors talk about normal/tumor/astrogliosis differentiation. None of the tissue samples were evaluated as normal brain. While no histology samples were taken from part of the brain with normal imaging, this doesn’t mean that histology would say the same thing. This has to be correctly defined throughout the article.
13.Have the authors thought of making a dynamic PET acquisition with proper pharmacokinetic modeling - how could this impact the performance of the method?
14.Several important articles from the field - e.g. 10.1093/neuonc/noz180 10.1007/s00259-023-06371-5 are missed.
Author Response
|
Response to Reviewer 3 Comments
|
||
|
1. Summary |
|
|
|
Thank you very much for taking the time to review this manuscript. Please find the detailed responses below and the corresponding revisions/corrections highlighted/in track changes in the re-submitted files. |
||
|
2. Point-by-point response to Comments and Suggestions for Authors |
|
|
|
Comments 1: Please provide basic demographics - number of patients, age+sex, which grades were included?
|
||
|
Response 1: Thank you for this suggestion, I have added demographics to the abstract. [Lines 31,32,33,34]
Comments 2 : Can you please explain what exactly this means? “increased sensitivity within FLAIR to”
Response 2: We rewrote this part of an abstract to make it clearer. Analyzing an optimal strategy for infiltration volume definition astrogliosis could be accurately differentiated from tumor samples using a choroid plexus as a background. Early acquisition improved the AUC in many cases, especially within FLAIR from 56% to 90% sensitivity and 41% to 61% specificity (standard TBR 1.6 vs early TBR plexus). [Line 39-42]
Comments 3. The results should be more focused and clear. Perhaps include less than more. 1), “although uptake in astrogliosis and grade 2 glioma was similar.” similar, but still allowing differentiation? 2) “to 71% but specificity remained low” - how low? 3) “Defining an optimal strategy for defining infiltration volume based on the choroid plexus, “ - unclear how is this related to the stated goals. Or is the choroid plexus used as background?
Response 3 See above. We have also clarified the aim of the study.
Therefore, the primary aim of this study was to differentiate astrogliosis and thalamus from tumor tissue, in order to optimize acquisition and analysis protocols for defining glioma infiltration especially within FLAIR and outside contrast-enhancement. To achieve this aim, we used not only dual time point acquisitions but also tested a variety of control tissues to obtain optimized thresholds. [lines 104-109]
METHODS/RESULTS 4. (MAJOR) Figure 1A - please adjust the image crop so that corresponding images cover a similar area. Also, make sure that any potentially sensitive information is removed - 1B is not really readable, but it is close. Response 4: Thank you for providing this valuable feedback. I have selected better image presenting exactly what the study is about. I hope that you also enjoyed, the tumor border sample is showing increases uptake in early and decreased uptake in late acquisition, which is localized inside FLAIR. (A) An image from biopsy planning system - A representative case of biopsy targeting (blue dot) different SUV at the tumor border in early (left) and late (middle) PET and FLAIR MRI (right) sequences
5. Figure 1D - Unclear difference between the two panels - explain that it’s the two different time points. The caption says that a median SUV is provided; why are there “THAMALUS” and “THALAMUS MAX”.?
Response 5 – Thank you for opinting this out, I have corrected the legend accordingly D) Median SUV values of different structures measured at both timepoints (early -left, standard-right), structures are presented from lowest to highest values; p-values indicate the closest differences that are significant. [Line154]
6. Line 109 - What was the brand of the scanner and MR field strength? TOF PET? Siemens Biograph mMR 3T, Unfortunately, this device does not have TOF and at a time when the study was performed there were no such devices. [Line 163-164]
7. Line 137-141 - Relevance of this paragraph is not entirely clear to me. Was this used for biopsy planning, then it can be perhaps shortened and moved way up.
Response 7: Thank you, I agree with your suggestion. I deleted this part as it was related to image fusion which is described above.
8. (MAJOR) Line 142 - What was the resolution of PET. In general, the resolution of PET is way too low to be used to check activity in intracranial arteries or even internal carotid arteries without facing partial volume effects, which complicates defining a reliable reference value. How was this dealt with? This might need to be listed as a limitation of this method. This also applies to samples from the tumor border - they all have lower SUV, but it is unclear if this is due to lower intake or simply because of partial volume effects on the thin peri-tumoral region.
Thank you for taking this point as it should be clarified. We have introduced some dedicated technical aspects to provide best quality of images. However, as you know that partial volume effects occur in many medical imaging methods, not only in PET. Please find below a detailed explanation that we also added to main text
The resolution (spatial resolution) of the Biograph mMR scanner declared by the scanner manufacturer is 4.6 mm in the central field of view (according to the NEMA standard for FBP, 344x344 matrix). In our imaging, a 344x344 matrix was used, to improve the resolution an iterative algorithm was additionally used, taking into account the angle of incidence of LOR on the detector block (HD PET, OSEM 3D + PSF) and appropriate statistics of counts were provided (acquisition duration 10 minutes). All corrections affecting the quality of imaging were also applied in the reconstruction. It can be assumed that the resolution in the study was less than 4 mm. All settings of the acquisition protocol were aimed at improving resolution and reducing the occurrence of undesirable phenomena such as partial volume effects.
9.Lines 163-178 - The paragraph needs more editing to make it clearer and easier to read.
Response 9: Unfortunately, this cannot be simplified. The methods given there are standard methods used to build classification trees. Excerpt “The global cross-validation costs were lower than the cross-validation costs for the selected tree, and similarly the standard error of the global cross-validation costs was close to the standard error of the selected tree costs, proving that the selected tree was appropriately sized and not overtrained. Furthermore, the cost of resubstitution was similar to the cost of cross-validation of the selected tree, confirming the selection of the best tree”. - is also a response to the query #11 about overtraining the tree
10. (MAJOR) Table 1 - Unclear what classification was really used when IDH-mutant glioblastomas are reported. Also, was 1p/19q used to classify oligodendrogliomas?
Response 10: Thank you for pointing this out as we have not changed it correctly according to new classification. Now it is corrected. 1p/19q codeletion we have tested in all but six patients as presented in supplementary table in our previous publication
https://static-content.springer.com/esm/art%3A10.1038%2Fs41467-023-39731-8/MediaObjects/41467_2023_39731_MOESM1_ESM.pdf
11. (MAJOR) Figure 2 - The decision tree with repeated use of the same parameter might be overtrained and spliting training and validation dataset is a must. Panels G and H are missing.
We use 3-fold cross-validation. The global costs of cross-validation are lower than the costs of cross-validation for the selected tree. Likewise, the standard error of the global cross-validation costs is close to the standard error of the costs of the selected tree. This proves that the selected tree is of the correct size and is not an over-trained tree. In addition, the cost of resubstitution is similar to the cost of a cross-validation for the selected tree, which is confirmed by the fact that it is the best tree. I hope that this explanation helps.
DISCUSSION:
12. (MAJOR) Line 325 - unclear why the authors talk about normal/tumor/astrogliosis differentiation. None of the tissue samples were evaluated as normal brain. While no histology samples were taken from part of the brain with normal imaging, this doesn’t mean that histology would say the same thing. This has to be correctly defined throughout the article.
Response 12: I fully agree with your valuable comment and suggestion. I have changed normal brain in to astrogliosis throughout the article as indeed we evaluated samples from radiologically affected tissue.
13.Have the authors thought of making a dynamic PET acquisition with proper pharmacokinetic modeling - how could this impact the performance of the method?
Response 13: Unfortunately, with this number of patients diagnosed in a busy department, performing full pharmacokinetic modeling is challenging. In our previous studies, we have shown that doing this at two timepoints could be sufficient. Nevertheless, in my opinion full modelling would even increase the precision as we could select the peak exactly.
Eg. doi: 10.1016/j.radonc.2016.06.004 or doi.org/10.1371/journal.pone.0140917
14.Several important articles from the field - e.g. 10.1093/neuonc/noz180 10.1007/s00259-023-06371-5 are missed. Response 14: I have added those papers to improve the discussion. Thank you. [line 559 and 573] |
||
Round 2
Reviewer 3 Report
Comments and Suggestions for Authors
- There's still patient 17 described as IDH-mutant glioblastoma
- Regarding the resolution and vessels - I agree that partial volume is a common problem for many modalities, and your methods for fixing the resolution seem appropriate. However, my comment was mostly directed at the ability to accurately extract the activity within the ICAs. Isn't the resolution borderline for this task, if yes, was the activity corrected for lesion diameter using vessel segmentation from, e.g., structural MRI. Or is this a limitation? Then list it in Discussion/Limitations.
- "T2_-weighted turbo 155 spin- echo or Gadalinium-Enhanced T1-weighted turbo spin-echo i" this is repeated three times in a row - perhaps some shortening is possible. Also please check the typo in "Gadalinium".
-
Author Response
Thank you again for taking your time to review our manuscript,
I have now corrected patient 17 diagnosis according to new classification
- Thank you for your insightful comment regarding resolution and vessel extraction. Indeed, partial volume effect poses a common challenge across various modalities, and our approach to addressing resolution issues is aligned with best practices.
Concerning the accuracy of extracting activity within the middle cerebral arteries, you raise a pertinent point. The resolution may indeed approach the borderline for such tasks. In our study, we did not employ activity correction based on lesion diameter using vessel segmentation from structural MRI. This could be considered a limitation of our methodology, particularly in the context of accurately capturing activity within MCAs.
We acknowledge this limitation and will include it in the Discussion section as part of the study's constraints. Thank you for bringing this aspect to our attention, and we value your contribution to enhancing the clarity and rigor of our research.
- "T2_-weighted turbo 155 spin- echo or Gadalinium-Enhanced T1-weighted turbo spin-echo i" this is repeated three times in a row - perhaps some shortening is possible. Also please check the typo in "Gadalinium".